# X-GRM: GAUSSIAN RECONSTRUCTION MODEL FOR SPARSE-VIEW X-RAYS TO COMPUTED TOMOGRAPHY

## ABSTRACT

Computed Tomography serves as an indispensable tool in clinical workflows, providing non-invasive visualization of internal anatomical structures. Existing CT reconstruction works are limited to small-capacity model architecture and inflexible volume representation. In this work, we present X-GRM (X-ray Gaussian Reconstruction Model), a large feedforward model for reconstructing 3D CT volumes from sparse-view 2D X-ray projections. X-GRM employs a scalable transformer-based architecture to encode sparse-view X-ray inputs, where tokens from different views are integrated efficiently. Then, these tokens are decoded into a novel volume representation, named Voxel-based Gaussian Splatting (VoxGS), which enables efficient CT volume extraction and differentiable X-ray rendering. This combination of a high-capacity model and flexible volume representation, empowers our model to produce high-quality reconstructions from various testing inputs, including in-domain and out-domain X-ray projections. Ours code and model weights will be available to the community.

## 1 INTRODUCTION

Computed Tomography (CT) serves as an indispensable tool in disease diagnosis, treatment planning, and intraoperative guidance (Cormack, 1963; Hounsfield, 1973; 1980). Traditional reconstruction methods (Feldkamp et al., 1984; Yu et al., 2006; Sidky & Pan, 2008a; Andersen & Kak, 1984) typically require hundred of X-ray projections to achieve diagnostically acceptable image quality, exposing patients to excessive radiation and also consuming substantial resources. To address these limitations, our work investigates reconstructing CT volumes from sparse-view X-ray projections.

Existing sparse-view CT reconstruction approaches can be categorized into optimization-based and regression-based methods. *Optimization-based methods* iteratively refine 3D volumes to match X-ray projections using neural representations (Zha et al., 2022; Shen et al., 2022; Zha et al., 2024b; Cai et al., 2024a;b) or diffusion models (Chung et al., 2023; 2024; Lee et al., 2023). While effective, these approaches typically require minutes to even hours for reconstructing per case, making them impractical for time-sensitive applications. *Regression-based methods* (Jin et al., 2017a) leverage neural networks to learn useful patterns for diverse reconstruction tasks-ranging from projection extrapolation (Anirudh et al., 2018; Ghani & Karl, 2018) and slice denoising (Wang et al., 2022; Jin et al., 2017b; Ma et al., 2023) to direct volume regression (Lin et al., 2023; 2024a;b). Despite enabling rapid inference, these approaches still face significant challenges: **(1)** they primarily employ small model architectures, like convolutional neural networks (CNNs), suffering from inherent scaling limitations, and **(2)** they primarily represent CT volumes as 3D voxels, which do not support differentiable X-ray rendering for incorporating X-ray constraints and render-related applications.

Very recently, X-LRM (Zhang et al., 2025a) introduced scalable transformer architectures trained on extensive datasets, while DeepSparse (Lin et al., 2025) explored similar large-scale approaches for CT reconstruction. Both methods achieve impressive reconstruction quality with computational efficiency. However, X-LRM relies on modified neural radiance fields (Zhang et al., 2025a) and DeepSparse uses discrete 3D voxel representations (Lin et al., 2025), neither of which support differentiable X-ray rendering. This limitation prevents these methods from incorporating direct X-ray constraints during training and restricts their applicability to rendering-based tasks.

To tackle these limitations, we propose a large feedforward X-ray Gaussian Reconstruction Model (**X-GRM**), for CT reconstruction from sparse-view X-ray projections. Our innovation is that a

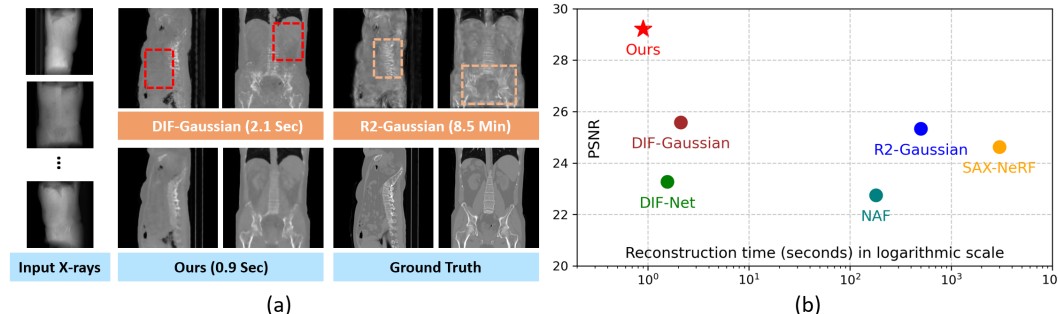

Figure 1: Our method achieves state-of-the-art reconstruction quality while maintaining the fastest runtime. (a) Qualitative results: DIF-Gaussian Lin et al. (2024a) exhibits issues with over-smooth results (red boxes) and R$^2$-Gaussian Zha et al. (2024a) has noise artifacts (orange boxes) and is time-consuming. In contrast, our method achieves better fidelity in a much shorter time. (b) Performance and runtime comparison: metrics are evaluated on the ReconX-15K test set.

transformer-based architecture (Dosovitskiy et al., 2021), when coupled with flexible volume representations, can excel at CT reconstruction. Specifically, we develop a scalable **X-ray Reconstruction Transformer** that employs an encoder ViT to efficiently tokenize each X-ray projection in parallel, followed by a fusion ViT that integrates global information by performing self-attention across views. Secondly, inspired by the efficient rendering of 3D Gaussian Splatting (Kerbl et al., 2023), we introduce a new volume representation, named **Voxel-based Gaussian Splatting** (VoxGS), where isometric 3D Gaussians are placed at voxel centers and rendering-related attributes are regressed from tokens. Such design enables efficient CT volume extraction and differentiable X-ray rendering, thereby introducing more constraints from X-ray inputs during training and also supports downstream render-related applications. As shown in Fig. 1, with the proposed scalable architecture and flexible volume representation, X-GRM drastically outperforms the SOTA methods in terms of both the reconstruction quality and inference speed.

In summary, the main contributions of our work are as follows:

- We introduce X-GRM, a large feed-forward Transformer model that, given sparse X-ray projection inputs, can directly predict the CT volume within one second.

- We introduce a flexible volume representation-Voxel-based Gaussian Splatting, which allows for efficient CT volume extraction and differentiable X-ray rendering.

- We demonstrate that X-GRM drastically outperforms existing methods in both the reconstruction quality and inference speed. Our codes and pre-trained models will be available.

## 2 RELATED WORK

### 2.1 SPARSE-VIEW CT RECONSTRUCTION

Sparse-view CT reconstruction tackles the complex task of creating complete 3D volumetric data from a limited number of X-ray projections. Traditional methods (Andersen & Kak, 1984; Sauer & Bouman, 1993; Sidky & Pan, 2008b) frame this as maximum a posteriori estimation with statistical priors, but often produce poor quality reconstructions when very few views are available. Recent advances have employed neural radiance fields (Zha et al., 2022; Shen et al., 2022; Cai et al., 2024b) or 3D Gaussian primitives (Cai et al., 2024a; Gao et al., 2024; Li et al., 2025; Zha et al., 2024b) as volume representations, which are gradually refined until their rendered X-rays match the input projections. Other approaches explore diffusion models as priors (Chung et al., 2023; 2024; Lee et al., 2023). While these methods deliver better results than traditional algorithms, they require extensive optimization time for each case, making them impractical for urgent clinical needs.

Feed-forward models offer a promising solution by directly mapping sparse projections to complete volumes with much faster inference times. These approaches include sinogram enhancement (Anirudh et al., 2018; Ghani & Karl, 2018), cross-sectional image denoising (Wang et al., 2022; Jin et al.,

2017b; Ma et al., 2023), and direct volume synthesis (Lin et al., 2023; 2024a;b). Though significantly faster, these methods are limited by their architectural design and rigid volume representations. Recent works like X-LRM (Zhang et al., 2025a) and DeepSparse (Lin et al., 2025) explore large foundation models for reconstruction, while our approach stands out through innovative architecture and 3D Gaussian volume representation–enabling both efficient reconstruction and high-quality X-ray rendering from any viewpoint, a capability missing in other approaches.

## 2.2 FEED-FORWARD 3D OBJECT/SCENE RECONSTRUCTION

Feed-forward reconstruction approaches aim to recover various 3D representations (e.g., mesh (Wang et al., 2018; Wu et al., 2020), implicit fields (Yu et al., 2021), 3DGS (Xu et al., 2024; Charatan et al., 2024; Chen et al., 2024), and point maps (Wang et al., 2024a)) through a single forward pass from input images. Specifically, in the domain of feed-forward object generation, LRM (Hong et al., 2024) and its variants (Li et al., 2024a; Tochilkin et al., 2024; Wang et al., 2023; Wei et al., 2024; Zhang et al., 2024; He et al., 2024; Cai et al., 2024c) have achieved significant advances in both performance and efficiency by leveraging large-scale datasets (Deitke et al., 2023b;a) and transformer architectures (Vaswani et al., 2017). Similarly, in feed-forward scene reconstruction, DUSt3R (Wang et al., 2024a), pixelSplat (Charatan et al., 2024) and subsequent variations (Leroy et al., 2024; Yang et al., 2025; Tang et al., 2024b; Zhang et al., 2025b; Wang et al., 2025b;a; Chen et al., 2024; Ye et al., 2024; Wang et al., 2024b; Tang et al., 2024a) have produced impressive results through continuous scaling up of their approaches. However, CT reconstruction remains in its infancy, lacking sufficient generalization capabilities due to the use of small-scale model architectures and inflexible volume representations. In this work, we aim to tackle these limitations.

## 3 METHOD

### 3.1 PRELIMINARIES

**CT imaging** forms the basis of computed tomography systems. When X-rays travel from source to detector, a projection $\boldsymbol{I} \in \mathbb{R}^{H \times W}$ captures the attenuation patterns through materials. For any ray $\boldsymbol{r}(t) = \boldsymbol{o} + t\boldsymbol{d} \in \mathbb{R}^3$ with source intensity $I_0$ traversing from distance $t_n$ to $t_f$, the detected intensity $I'(\boldsymbol{r})$ adheres to the Beer-Lambert principle: $I'(\boldsymbol{r}) = I_0 \exp(-\int_{t_n}^{t_f} \sigma(\boldsymbol{r}(t))dt)$, with $\sigma(\boldsymbol{x})$ denoting the volumetric density at position $\boldsymbol{x} \in \mathbb{R}^3$. For numerical stability and analytical convenience, raw measurements undergo logarithmic transformation: $I(r) = \log I_0 - \log I'(\boldsymbol{r}) = \int_{t_n}^{t_f} \sigma(\boldsymbol{r}(t))dt$. This converts each measurement into a line integral of material density along the ray trajectory. The fundamental challenge in CT reconstruction involves recovering the underlying 3D density field $\sigma(\boldsymbol{x})$ from a series of projections $\{\boldsymbol{I}_i\}_{i=1}^{K}$ acquired across $K$ distinct angular positions.

**3D Gaussian Splatting** Kerbl et al. (2023) represents an approach to 3D content modeling through a collection of $N_p$ colored Gaussian primitives $\mathcal{G} = \{\boldsymbol{g}_i\}_{i=1}^{N_p}$. Each Gaussian element $\boldsymbol{g}_i = \exp\left(-\frac{1}{2}(\boldsymbol{x} - \boldsymbol{\mu}_i)^\top \boldsymbol{\Sigma}_i^{-1}(\boldsymbol{x} - \boldsymbol{\mu}_i)\right)$ is characterized by its opacity $\boldsymbol{\sigma}_i \in \mathbb{R}$ and color $\boldsymbol{c}_i \in \mathbb{R}^3$ properties, alongside positional parameters $\boldsymbol{\mu}_i \in \mathbb{R}^3$ and covariance matrix $\boldsymbol{\Sigma}_i \in \mathbb{R}^{3 \times 3}$ that encodes scaling $\boldsymbol{s}_i \in \mathbb{R}^3$ and orientation quaternion $\boldsymbol{r}_i \in \mathbb{R}^4$ within 3D space. This comprehensive set of attributes forms $\boldsymbol{G} \in \mathbb{R}^{N_p \times 14}$, capable of encoding complex 3D scenes or objects. By aligning the formula, the rasterization process is equivalent to the CT imaging procedure, thus we can use 3DGS to represent CT volumes and render X-rays (see Sec. A.1 for more details). Our approach also modify the original 3DGS to address the distinct requirements of feed-forward CT reconstruction tasks.

### 3.2 OVERVIEW

Given a set of sparse-view X-ray projections $\mathcal{I} = \{\boldsymbol{I}_i\}_{i=1}^{K}$, $(\boldsymbol{I}_i \in \mathbb{R}^{H \times W})$ and their corresponding projection matrix $\mathcal{P} = \{\boldsymbol{P}_i\}_{i=1}^{K}$, $\boldsymbol{P}_i = \boldsymbol{K}_i[\boldsymbol{R}_i|\boldsymbol{t}_i]$, calculated from via intrinsics matrix $\boldsymbol{K}_i$, rotation matrix $\boldsymbol{R}_i$, and translation vector $\boldsymbol{t}_i$, our goal is to learn a mapping $f_{\boldsymbol{\theta}}$ from X-ray projections to CT volume density field $\boldsymbol{V} \in \mathbb{R}^{M \times N \times L}$:

$$f_{\boldsymbol{\theta}} : \{\boldsymbol{I}_i, \boldsymbol{P}_i\}_{i=1}^{K} \mapsto \{\boldsymbol{V}(x, y, z) | x \in [1, M], y \in [1, N], z \in [1, L]\}, \quad (1)$$

where we formulate $f_{\boldsymbol{\theta}}$ as a feedforward transformer and $\boldsymbol{\theta}$ are learnable parameters optimized from a large-scale training dataset. As illustrated in Fig. 2, our method consists of a large X-ray

Figure 2: **X-GRM** is a large feed-forward transformer trained on a curated large CT reconstruction dataset. (a) X-ray Reconstruction Transformer efficiently encodes and fuses tokens from multiple X-ray projections, and (b) Voxel-based Gaussian Splatting enables both the efficient CT volume extraction and differentiable X-ray rendering.

Reconstruction Transformer to enhance the model scalability and a flexible voxel-based Gaussian Splatting that enables efficient CT volume extraction and differentiable rendering. In this section, we first describe the process of encoding X-ray inputs into compact latent tokens, integrated with camera information (Sec. 3.3). Then, we discuss the design of VoxGS and how to regress Gaussian attributes from encoded tokens (Sec. 3.4). Finally, we describe the model training details (Sec. 3.5).

## 3.3 X-RAY RECONSTRUCTION TRANSFORMER

Existing CNN-based CT reconstruction methods (Lin et al., 2023; 2024a;b) face inherent scaling limitations that restrict their effectiveness for complex volumetric reconstruction. To overcome these constraints, we propose a large X-ray Reconstruction Transformer that significantly enhances CT reconstruction capabilities. Our approach employs an encoder ViT to efficiently tokenize X-ray projections in parallel, followed by a fusion ViT that integrates tokens across multiple views to reconstruct high-quality 3D volumes. Additionally, we incorporate camera geometry information into tokens through ModLN (Peebles & Xie, 2023), enabling more precise spatial reasoning.

**Encoder ViT.** We encode each X-ray projection $\boldsymbol{I}_i \in \mathcal{I}$ to a set of patch features $\boldsymbol{H}_i$, using a feature extractor $\mathcal{F}_{enc}$. This is done independently per image, yielding a sequence of image patch features $\boldsymbol{H}_i = \{h_{i,j}\}_{i,j=1}^{HW/P^2}$ for each image:

$$\boldsymbol{H}_i = \mathcal{F}_{enc}(\boldsymbol{I}_i), i = 1, ..., K. \tag{2}$$

Here we adopt transformer-based DINO (Zhang et al., 2022) as our encoder. Before passing image features $\{\boldsymbol{H}_i\}_{i=1}^K$ to the decoder, we also inject Plücker ray directions into the features via adaptive layer norm (Peebles & Xie, 2023):

$$\tilde{\boldsymbol{H}}_i = ModLN(\boldsymbol{H}_i, \boldsymbol{R}_i), i = 1, ..., K, \tag{3}$$

where $\boldsymbol{R}_i$ is the Plücker ray embedding of the camera angle and origin. Unlike previous works that modulate camera poses to image features using extrinsic and intrinsic matrices (Hong et al., 2024; Liu et al., 2023a), Plücker rays are defined by the cross product between the camera location and ray direction, offering a unique ray parameterization independent of scale, camera position and focal length.

**Fusion ViT.** Most of the computation in our framework happens in the fusion ViT. We use a 16-layer transformer similar to ViT-B (Dosovitskiy et al., 2021). This fusion transformer takes the concatenated encoded image patches from all views $\{\tilde{\boldsymbol{H}}_i\}_{i=1}^K$ and performs all-to-all self-attention $\mathcal{F}_{fuse}$:

$$\{\boldsymbol{F}_1, \boldsymbol{F}_2, ..., \boldsymbol{F}_K\} = \mathcal{F}_{fuse}(\tilde{\boldsymbol{H}}_1, \tilde{\boldsymbol{H}}_2, ..., \tilde{\boldsymbol{H}}_K), \tag{4}$$

where $\boldsymbol{F}_i$ is the fused token of $i$-th view. This operation provides features of each view with full context from all other views, enables complete spatial and context reasoning.

## 3.4 Voxel-based Gaussian Splatting

Existing feedforward models (Lin et al., 2024a) typically represent CT as 3D voxels, yet it has limitations of supporting differentiable real-time rendering, thus can hardly leverage the X-ray projection constraints during the training period. Benefiting from highly optimized rasterization, 3DGS (Kerbl et al., 2023) enables real-time differentiable rendering. However, directly representing CT with original 3DGS or modified 3DGS for CT (Zha et al., 2024a; Gao et al., 2024) leads to inaccurate volume extraction and instable optimization for feedforward framework (Tab. 6). Therefore, we propose a new CT representation-Voxel-based Gaussian Splatting (VoxGS) to support precise volume extraction and differentiable X-ray rendering.

**Voxel-based Gaussian Splatting.** VoxGS has two different designs with the original 3DGS. *Firstly*, it consists of 3D Gaussians $\mathcal{G} = \{\boldsymbol{g}_i\}_{i=1}^{N_p}$ with fixed positions. In particular, the position $\boldsymbol{\mu}_i$ of each Gaussian $\boldsymbol{g}_i$ is fixed at the centroid of each voxel grid $\boldsymbol{V}(x, y, z)$:

$$\boldsymbol{\mu}_i = (x, y, z), x \in [1, M], y \in [1, N], z \in [1, L]. \tag{5}$$

This design offers the advantage that when extracting CT volumes from 3D Gaussians $\mathcal{G}$, complex trilinear interpolation strategies are not required for identifying the voxel value, instead we can directly obtain it from the Gaussians opacities, which improves extraction speed and enables more stable optimization. *Secondly*, we remove the SH coefficients $\boldsymbol{c}_i$ related to color rendering from the Gaussian attributes, retaining only the opacity $\alpha_i$, the scale vector $\boldsymbol{s}_i$, and the rotation vector $\boldsymbol{r}_i$. This design is similar to (Zha et al., 2024a), where the kernel formulation removes view-dependent color because X-ray attenuation depends only on isotropic density. In summary, we can represent a CT volume as a set of voxel-based Gaussians: $\mathcal{G} = \{\boldsymbol{g}_i | \boldsymbol{g}_i = \{\boldsymbol{\mu}_i, \alpha_i, \boldsymbol{s}_i, \boldsymbol{r}_i\}, \boldsymbol{\mu}_i \text{ is fixed}\}_{i=1}^{N_p}$, from which X-ray projections can be efficiently rendered in a differentiable manner.

**Decoding VoxGS attributes.** As Gaussian positions $\boldsymbol{\mu}_i$ is fixed, we only need to decode Gaussians attributes $\alpha_i$ and $\boldsymbol{\Sigma}_i$ from available view features $\{\boldsymbol{F}_1, \boldsymbol{F}_2, ..., \boldsymbol{F}_K\} \subset \mathbb{R}^{H \times W \times C}$, where $K$ is the number of X-ray projections. Specifically, for a 3D Gaussian $\boldsymbol{g}_i$ with fixed position $\boldsymbol{\mu}_i$, we query view-specific features $\tilde{\boldsymbol{F}}_k$ from $\boldsymbol{F}_k$:

$$\boldsymbol{F}_k^i = Interp(\boldsymbol{F}_k, \boldsymbol{P}_k(\boldsymbol{\mu}_i)) \in \mathbb{R}^C, \text{ for } k = 1, 2, ..., K, \tag{6}$$

where $\boldsymbol{P}_k : \mathbb{R}^3 \to \mathbb{R}^2$ is the projection matrix of $k$-th view, and $Interp(\cdot)$ denotes bilinear interpolation. Then, $K$ queried features are aggregated by a MaxPooling layer to obtain $\boldsymbol{F}_{max}^i = MaxPool(\boldsymbol{F}_1^i, \boldsymbol{F}_2^i, ..., \boldsymbol{F}_K^i) \in \mathbb{R}^C$. Finally, $\boldsymbol{F}_{max}^i$ is passed to several MLP layers to predict Gaussian parameters:

$$[\alpha_i, \boldsymbol{s}_i, \boldsymbol{r}_i] = MLPs(\boldsymbol{F}_{max}^i) \in \mathbb{R}^{1+3+4}. \tag{7}$$

**Volume extraction and X-ray rendering.** To extract the CT density volume $\boldsymbol{V} \in \mathbb{R}^{M \times N \times L}$, we can directly index the corresponding fixed Gaussian opacities $\alpha_i$:

$$\boldsymbol{V}(x, y, z) = \alpha_i, \text{ where } \boldsymbol{\mu_i} = (x, y, z)^T. \tag{8}$$

Benefiting from the fixed position design of VoxGS, extracting the density field can be achieved through fast indexing operations, without need of extra computation. While for X-ray rendering, we employ the rasterizer implemented in (Zha et al., 2024a) to render projections in a differentiable manner, which enables more constraints from inputs X-rays and also support various downstream applications, like novel view synthesis and potential CT/X-ray registration.

## 3.5 Training objectives and protocols

**Training objectives.** We train the network with two objectives: 1) to impose constraints on the predicted CT volume $\boldsymbol{V}$ such that it aligns with the ground truth $\boldsymbol{V}_{gt}$ and 2) to minimize the difference between the actual X-ray projections $\boldsymbol{I}_{gt}$ and rendered ones $\boldsymbol{I}$. Accordingly, we adopt two categories of losses: volume constraints and rendering constraints: $\mathcal{L} = \mathcal{L}_{volume} + \mathcal{L}_{render}$. Specifically, for volume reconstruction constraint $\mathcal{L}_{volume}$, we adopt MSE to compute point-wise estimation error:

$$\mathcal{L}_{volume}(\boldsymbol{V}, \boldsymbol{V}_{gt}) = \sum_{x,y,z} (\boldsymbol{V}(x, y, z) - \boldsymbol{V}_{gt}(x, y, z))^2. \tag{9}$$

While for rendering constraint $\mathcal{L}_{render}$, we use a weight combination of photometric L1 loss $\mathcal{L}_1$ and D-SSIM loss $\mathcal{L}_{ssim}$ (Wang et al., 2004):

$$\mathcal{L}_{render}(\boldsymbol{I}, \boldsymbol{I}_{gt}) = \lambda_{L1}\mathcal{L}_1(\boldsymbol{I}, \boldsymbol{I}_{gt}) + \lambda_{ssim}\mathcal{L}_{ssim}(\boldsymbol{I}, \boldsymbol{I}_{gt}). \tag{10}$$

**Training protocols.** During the training period, we observe rendering with the original volume resolution $M \times N \times K$ ($256^3$ in this work), requires exhaustive GPU memory to preserve gradients. Therefore, we instead randomly sample sub-volumes $\tilde{V} \in \mathbb{R}^{M/4 \times N/4 \times K/4}$ from the original volume $V$, and modify the training objectives of Eq. 9 accordingly. While during the inference, we directly use the full volume $V$, as no gradients are required to store.

## 4 EXPERIMENT

### 4.1 EXPERIMENTAL SETUP

**Datasets.** To support the training of our scalable transformer with diverse anatomical data, we assemble an extensive dataset following X-LRM's (Zhang et al., 2025a) selection, combining 14,972 CT volumes and their corresponding X-ray projections from 8 public datasets (detailed in Tab.1). This comprehensive repository spans the most clinically relevant body regions, including chest, abdomen, pelvis, and dental structures. For systematic model development and evaluation, we divided the collection into three distinct subsets: 13,612 volumes for training, 680 for validation, and 680 for testing. Complete specifications of these dataset partitions are thoroughly documented in Sec.A.2.

Table 1: The statistics of collected datasets.

| Dataset | Body Regions | # Volumes |
|---|---|---|
| AbdomenAtlas v1.0 (Li et al., 2024b) | Abdomen, Chest, Pelvis | 5,171 |
| RSNA2023 (Hermans et al., 2024) | Abdomen, Pelvis | 4,711 |
| LUNA16 (Setio et al., 2017) | Chest | 833 |
| AMOS (Ji et al., 2022) | Abdomen | 1,851 |
| MELA (Organizers, 2022) | Chest | 1,100 |
| RibFrac (Jin et al., 2020; Yang et al., 2024) | Abdomen, Chest | 660 |
| ToothFairy2 (Bolelli et al., 2024) | Tooth | 223 |
| STSTooth (wang, 2024) | Tooth | 423 |
| In total | All above | 14,972 |

For CT standardization, we meticulously normalize anatomical regions: chest, abdomen, and pelvic volumes are resampled to $50^3\text{cm}^3$ at $256^3$ voxel resolution, while dental acquisitions are calibrated to $40^3\text{cm}^3$ with the same resolution. We also transform radio-density measurements from native Hounsfield units (chest, abdomen, pelvis: [-1000, 1000]; dental: [-1000, 3000]) to a unified [0,1] scale, preserving crucial diagnostic information across key structures. To render X-ray projections from CT volumes in each dataset, we employ the TIGRE (Biguri et al., 2016) toolbox to generate synthetic radiographic projections—producing 50 high-definition views ($256^2$ resolution) distributed uniformly throughout the complete angular spectrum ($[0°, 360°]$). Clinical authenticity was enhanced through noise integration, combining Gaussian and Poisson distributions to accurately replicate quantum photon effects and Compton scatter phenomena.

**Implementation details.** The proposed X-GRM is implemented with PyTorch and trained on 4 NVIDIA A100 (40G) GPUs. The model is optimized with AdamW (Loshchilov & Hutter, 2017) for 100 epochs, with a batch size of 8 and $\beta_1 = 0.9, \beta_2 = 0.95$. The initial learning rate is set to $1 \times 10^{-4}$ and gradually decreases to $1 \times 10^{-6}$ following the cosine annealing scheduler (Loshchilov & Hutter, 2016). For the model implementation, the encoder ViT $\mathcal{F}_{enc}$ uses a ViT-B/16 (Dosovitskiy et al., 2021) architecture, initialized from DINO (Zhang et al., 2022) pre-trained weights. The fusion ViT $\mathcal{F}_{fuse}$ is a ViT-B/16 model initialized from scratch, which has $16 \times 16$ patch size, 12 layers, 12 heads, embedding dimension 768, and MLP ratio 4.0. We leverage bFloat16 precision and gradient checkpointing to improve GPU memory and computational efficiency.

During model training, we employ a variable-view learning strategy with the number of views randomly selected from $6, 8, 10$, with these perspectives evenly sampled from the 50 available projections. For quantitative evaluation, we utilize complementary metrics: PSNR calculated directly within the 3D volumetric space provides assessment of signal fidelity, while SSIM averaged across multiple 2D slice comparisons offers insight into structural preservation, together providing a comprehensive measurement of reconstruction quality.

Table 2: Quantitative comparison with traditional and feedforward methods. We evaluate all methods using the full test set ($256^3$ resolution) on one A100 GPU. All feedforward models are trained on the same training set. **Bold**: best results; underlined: second-best results.

| Method | Time (s)↓ | 6-View | | 8-View | | 10-View | |
|---|---|---|---|---|---|---|---|
| | | PSNR↑ | SSIM↑ | PSNR↑ | SSIM↑ | PSNR↑ | SSIM↑ |
| FDK (Feldkamp et al., 1984) | **0.297** | 14.65 | 0.297 | 15.50 | 0.295 | 16.23 | 0.314 |
| ASD-POCS (Sidky & Pan, 2008a) | 5.704 | 21.47 | 0.498 | 22.32 | 0.560 | 22.62 | 0.653 |
| SART (Andersen & Kak, 1984) | 21.091 | 21.89 | 0.634 | 22.52 | 0.642 | 23.66 | 0.685 |
| FBPConvNet (Jin et al., 2017a) | 0.131 | 22.63 | 0.691 | 23.05 | 0.695 | 23.48 | 0.718 |
| FreeSeed (Ma et al., 2023) | 0.508 | 24.67 | 0.781 | 24.89 | 0.789 | 25.43 | 0.790 |
| DIF-Net (Lin et al., 2023) | 1.561 | 22.27 | 0.684 | 22.72 | 0.688 | 23.27 | 0.714 |
| DIF-Gaussian (Lin et al., 2024a) | 2.120 | 24.83 | 0.787 | 25.03 | 0.793 | 25.58 | 0.794 |
| **X-GRM (Ours)** | 0.928 | **28.39** | **0.873** | **28.86** | **0.879** | **29.21** | **0.886** |

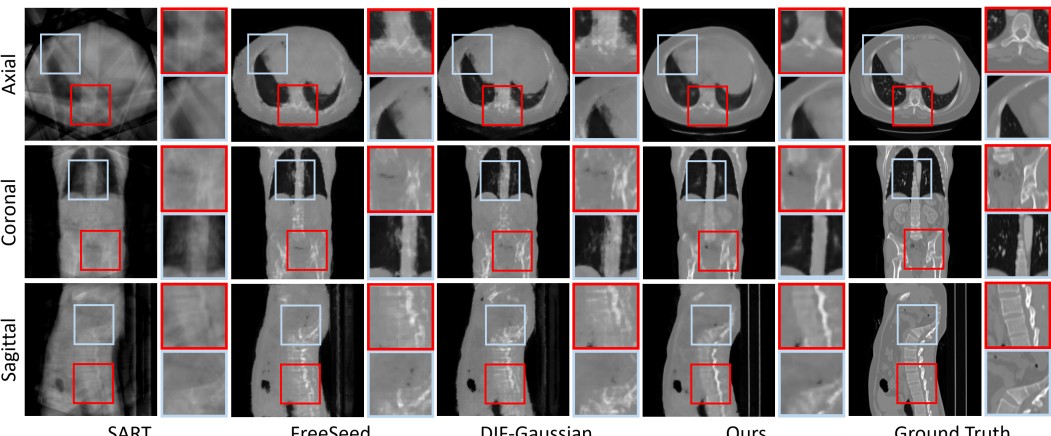

Figure 3: Qualitative comparison with traditional and feedforward methods. Results shown are from the test set reconstructions with 10-view inputs.

## 4.2 ASSESSING CT RECONSTRUCTION

Following the setup of X-LRM (Zhang et al., 2025a), we evaluate our X-GRM against existing CT reconstruction methods with two settings. Setting 1: evaluating on the full test set (680 samples) of the collected dataset. This setting includes traditional and feedforward methods, as they can efficiently handle massive X-ray inputs. To ensure fair comparison, we re-train the compared feedforward models on the same training set. Setting 2: evaluating on a sampled set (40 samples). This setting includes self-supervised approaches, as they typically require substantial optimization time. To cover diverse organs, we evenly sample five volumes from the test set of eight datasets.

**Setting 1.** We first compare with traditional methods (FDK (Feldkamp et al., 1984), SART (Andersen & Kak, 1984), ASD-POCS (Sidky & Pan, 2008a)) and feedforward methods (FBPConvNet (Jin et al., 2017a), FreeSeed (Ma et al., 2023), DIF-Net (Lin et al., 2023), DIF-Gaussian (Lin et al., 2024a)). As shown in Tab.2, X-GRM outperforms the compared models in both reconstruction quality and inference speed. When compared to FreeSeed, the SOTA 2D feedforward model, X-GRM achieves substantial improvements of 3.71dB, 3.97dB, and 3.78dB in PSNR for 6-view, 8-view, and 10-view inputs, respectively. Similarly, X-GRM outperforms DIF-Gaussian, the leading 3D feedforward method, by 3.56dB, 3.83dB, and 3.63dB while delivering $2\times$ inference speed. Some qualitative results are presented in Fig.4 (more results in Sec. A.6). Traditional methods like SART introduce prominent streak artifacts in sparse-view reconstructions. Meanwhile, feedforward approaches like FreeSeed and DIF-Gaussian suffer from significant blurring, particularly in bone structures and regions with low tissue contrast. In contrast, our proposed X-GRM method can preserve fine anatomical details while maintaining structural integrity.

**Setting 2.** We then compare with self-supervised methods (NAF (Zha et al., 2022), SAX-NeRF (Cai et al., 2024b), $R^2$-Gaussian (Zha et al., 2022)). As shown in Tab.8, X-GRM has clear advantages over

Table 3: Quantitative comparison with self-supervised methods. We evaluate all methods using 40 CT volumes ($256^3$ resolution) sampled from the full test set on one RTX 4090Ti GPU. **Bold** and underlined values indicate best and second-best results.

| Method | Time↓ | 6-View | | 8-View | | 10-View | |
|---|---|---|---|---|---|---|---|
| | | PSNR↑ | SSIM↑ | PSNR↑ | SSIM↑ | PSNR↑ | SSIM↑ |
| NAF (Zha et al., 2022) | 3.0m | 20.69 | 0.530 | 21.93 | 0.563 | 22.75 | 0.581 |
| SAX-NeRF (Cai et al., 2024b) | 48.5m | 22.93 | 0.686 | 23.82 | 0.702 | 24.62 | 0.730 |
| $R^2$-Gaussian (Zha et al., 2024a) | 8.5m | 22.90 | 0.662 | 24.09 | 0.689 | 25.33 | 0.727 |
| **X-GRM (Ours)** | **0.9s** | **27.71** | **0.838** | **28.16** | **0.846** | **28.47** | **0.854** |

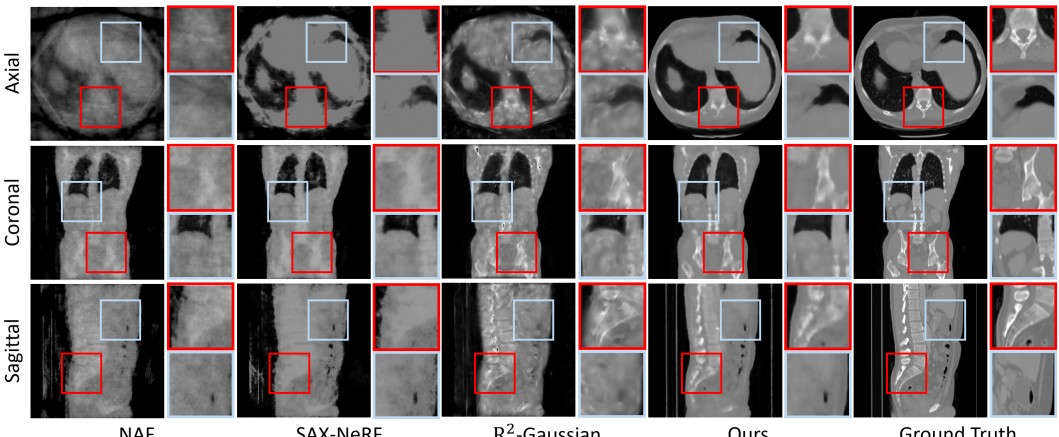

Figure 4: Qualitative comparison with self-supervised models. Results shown are from the test set reconstructions with 10-view inputs.

existing self-supervised methods: it achieves the best reconstruction performance while requiring the least computational time. Compared to the second-fastest self-supervised method NAF, X-GRM is 200× faster, requiring only 0.9 second to reconstruct one CT volume of resolution $256^3$. While compared to the state-of-the-art method SAX-NeRF and $R^2$-Gaussian, X-GRM delivers 4.78/4.34/3.85dB and 4.81/4.01/3.14dB higher PSNR when processing 6/8/10 inputs views. As evident from the reconstructed slices in Fig.4 (more results in Sec. A.6), NAF exhibit noise artifacts, SAX-NeRF produces less noise, but many anatomical structures are excessively smoothed, $R^2$-Gaussian better recovers anatomical structures, but exhibits many discontinuous regions, likely due to floating artifacts from 3DGS. In contrast, the reconstructed slices from X-GRM exhibit clearer anatomical structures, more accurate bone density, and fewer artifacts throughout the volume slices.

## 4.3 ASSESSING CROSS-DATASET GENERALIZATION

Due to the use of scalable architecture and large-scale training, X-GRM is inherently superior in generalizing to *out-of-distribution* inputs. To demonstrate this advantage, we conduct two cross-dataset evaluations. In particular, we directly test the model on unseen PENGWIN (Liu et al., 2023b) (pelvis) and FUMPE (Masoudi et al., 2018) (chest) datasets.

As evident in Tab. 4, our X-GRM significantly outperforms other feedforward models, including FreeSeed, DIF-Net, and DIF-Gaussian. Compared to self-supervised models that uses per-sample optimization, our method also shows comparable or better performance, while delivering much faster inference speed (around 500× faster compared to $R^2$-Gaussian).

Table 4: Cross-dataset reconstruction evaluation.

| Method | Time | FUMPE | | PENGWIN | |
|---|---|---|---|---|---|
| | | PSNR | SSIM | PSNR | SSIM |
| FreeSeed (Ma et al., 2023) | **0.5s** | 22.69 | 0.760 | 25.09 | 0.824 |
| DIF-Net (Lin et al., 2023) | 1.5s | 20.49 | 0.640 | 23.72 | 0.724 |
| DIF-Gaussian (Lin et al., 2024a) | 2.0s | 23.50 | 0.768 | 25.46 | 0.827 |
| SAX-NeRF (Cai et al., 2024b) | 48.5m | 23.58 | 0.722 | 24.72 | 0.745 |
| $R^2$-Gaussian (Zha et al., 2024a) | 8.6m | 23.09 | 0.679 | **26.19** | 0.782 |
| **X-GRM (Ours)** | 1.0s | **24.81** | **0.811** | 26.04 | **0.856** |

Table 5: Novel view synthesis results. Time: rendering time of one X-ray projection.

| Method | PSNR ↑ | SSIM ↑ | Time(s) ↓ |
|---|---|---|---|
| NAF | 28.29 | 0.505 | 0.448 |
| SAX-NeRF | 31.85 | 0.659 | 0.916 |
| $R^2$-Gaussian | 38.26 | 0.955 | **0.003** |
| **X-GRM (Ours)** | **49.44** | **0.993** | 0.020 |

Figure 5: Qualitative results of novel views.

Table 6: Ablation study of component designs on the ReconX-16K dataset with 10-view inputs.

| (a) Camera pose integration. | | | (b) Volume representation. | | | (c) Cross-view aggregation. | | |
|---|---|---|---|---|---|---|---|---|
| Method | PSNR↑ | SSIM↑ | Method | PSNR↑ | SSIM↑ | Method | PSNR↑ | SSIM↑ |
| w/o pose | 28.93 | 0.862 | w/o VoxGS | 28.66 | 0.858 | w/o attention | 28.79 | 0.864 |
| dense add | 29.15 | 0.882 | VoxGS w shift | 26.84 | 0.837 | cross-attention | 29.18 | 0.882 |
| **Ours** (ModLN) | **29.21** | **0.886** | **Ours** (VoxGS) | **29.21** | **0.886** | **Ours** (Self-attn) | **29.21** | **0.886** |

## 4.4 ASSESSING X-RAY SYNTHESIS

We evaluate X-GRM against established self-supervised methods that support X-ray synthesis using NeRF or 3DGS representations. Our evaluation is conducted on the test set comprising 30 distinct CT samples. For each sample, we use 10 projection views as input, and evaluate using the rest 40 unseen novel views. As reported in Tab. 5, X-GRM delivers 12.55dB higher PSNR over $R^2$-Gaussian while maintaining fast rendering speeds of 0.02s for each projection. Qualitative analysis in Fig. 5 shows that SAX-NeRF and $R^2$-Gaussian generate substantial noise artifacts at tissue-air interfaces (red boxes), whereas X-GRM consistently preserves clear, well-defined anatomical boundaries.

## 4.5 ABLATION STUDY

**Camera pose integration.** We evaluate two variants: removing pose integration (w/o pose') and directly adding ray embeddings to encoded tokens (dense add'). As shown in Tab. 6(a), removing camera pose decreases PSNR by 0.28dB, confirming its necessity for CT geometry reasoning. While 'dense add' slightly underperforms ModLN, we adopt ModLN for its superior performance.

**Volume representation.** We compare using voxel grids directly (w/o VoxGS') against introducing shift attributes for Gaussian position prediction (VoxGS w/ shift'). Tab. 6(b) shows that w/o VoxGS' reduces PSNR by 0.55dB, demonstrating the benefits of Gaussian rendering constraints. Notably, VoxGS w/ shift' suffers a 2.37dB drop, likely due to challenges in optimizing precise Gaussian positions and volume extraction inaccuracies.

**Cross-view aggregation.** We test removing the fusion ViT (w/o attention') and replacing self-attention with cross-attention (cross-attention'). Tab. 6(c) reveals that eliminating attention decreases PSNR by 0.52dB, validating the critical importance of cross-view information aggregation for accurate geometric reasoning. Since 'cross-attention' achieves comparable results to self-attention, we choose the latter for its simpler implementation.

## 5 CONCLUSION

This paper presents X-GRM, a large feedforward model for reconstructing 3D CT volumes from sparse-view X-rays using a scalable transformer architecture and Voxel-based Gaussian Splatting representation. Trained on a large CT reconstruction dataset, X-GRM demonstrates superior reconstruction quality with sparse-view X-ray inputs. By addressing previous limitations in model capacity and volume representation, our work represents an advancement in non-invasive sparse-view CT imaging for clinical applications.

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

# A    TECHNICAL APPENDICES AND SUPPLEMENTARY MATERIAL

## A.1    REPRESENT CT VOLUMES AS 3D GAUSSIANS

In 3D Gaussian Splatting (3DGS) methods (Kerbl et al., 2023) for natural scene rendering, the attenuation of light as it passes through a medium is described using an exponential transmittance model $T(t)$. This model is physically based on the Beer-Lambert law (Vicini et al., 2021), representing the light propagation process from one point $r(t_0)$ to another point $r(t)$ in space:

$$T(t) = \exp(-\int_{t_0}^{t} \sigma(r(s))ds). \tag{11}$$

Here, $r(t) = o + td$ represents a 3D point along a ray with origin $o$ and direction $d$, while $\sigma(p)$ denotes the density at point $p$. As established in Sec. 3.1, X-ray attenuation follows an identical model. Consequently, 3DGS rendering can be directly leveraged for X-ray projection rasterization under a simplified imaging model that accounts solely for isotropic absorption (per Beer-Lambert law), with optional exclusion of color attributes.

## A.2    DATASET DETAILS

As stated in the main text, we follow X-LRM (Zhang et al., 2025a) to collect 8 public datasets covering organs and structures including chest, abdomen, pelvis, and teeth. We apply data standardization to adjust the data through three steps: resampling spacing, adjusting resolution to $256^3$, clipping HU values, and normalizing to $[0, 1]$. Then, we split each dataset into training/validation/test splits with ratio 20/1/1. Meanwhile, as most datasets solely contain CT volumes, we utilize the TIGRE toolbox (Biguri et al., 2016) to render X-ray projections. The specific parameter settings for each dataset are shown in the Tab. 7.

Table 7: The parameters used to pre-process each subset of the collected dataset.

| Dataset | # Volumes | Train/Val/Test | Spacing (mm) | Clipping range |
|---|---|---|---|---|
| AbdomenAtlas v1.0 (Li et al., 2024b) | 5,171 | 4701/235/235 | [1.0,1.0,1.0] | [-1000, 1000] |
| RSNA2023 (Hermans et al., 2024) | 4,711 | 4283/214/214 | [1.0,1.0,1.0] | [-1000, 1000] |
| LUNA16 (Setio et al., 2017) | 833 | 757/38/38 | [1.0,1.0,1.0] | [-1000, 1000] |
| AMOS (Ji et al., 2022) | 1,851 | 1683/84/84 | [1.0,1.0,1.0] | [-1000, 1000] |
| MELA (Organizers, 2022) | 1,100 | 1000/50/50 | [1.0,1.0,1.0] | [-1000, 1000] |
| RibFrac (Jin et al., 2020; Yang et al., 2024) | 660 | 600/30/30 | [1.0,1.0,1.0] | [-1000, 1000] |
| ToothFairy2 (Bolelli et al., 2024) | 223 | 203/10/10 | [0.3,0.3,0.3] | [-1000, 2000] |
| STSTooth (wang, 2024) | 423 | 385/19/19 | [0.3,0.3,0.3] | [-1000, 2000] |

## A.3    ADDITIONAL EXPERIMENTS OF NOVEL VIEW SYNTHESIS

Due to space constraints, Tab. 5 only presents results for novel view synthesis with 10 input views. Here, we further compare performance with 6 and 8 input views, as shown in the table. Our method consistently outperforms existing self-supervised approaches by a significant margin. In particular, compared to the current best method $R^2$-Gaussian, our approach delivers 14.92/13.01/11.18 dB improvement in performance with 6/8/10 input views, respectively. Meanwhile, our method achieves a rendering speed of 0.02s for each projection, meeting the requirements for most scenarios.

Table 8: Additional experiments of novel view synthesis on the sampled test set (40 samples).

| Method | Render Time (s)↓ | 6-View | | 8-View | | 10-View | |
|---|---|---|---|---|---|---|---|
| | | PSNR↑ | SSIM↑ | PSNR↑ | SSIM↑ | PSNR↑ | SSIM↑ |
| NAF (Zha et al., 2022) | 0.448 | 23.11 | 0.404 | 26.07 | 0.463 | 28.29 | 0.505 |
| SAX-NeRF (Cai et al., 2024b) | 0.916 | 27.72 | 0.562 | 29.09 | 0.603 | 31.85 | 0.659 |
| $R^2$-Gaussian (Zha et al., 2024a) | **0.003** | 34.23 | 0.912 | 36.33 | 0.930 | 38.26 | 0.955 |
| **X-GRM (Ours)** | 0.020 | **49.15** | **0.991** | **49.34** | **0.992** | **49.44** | **0.993** |

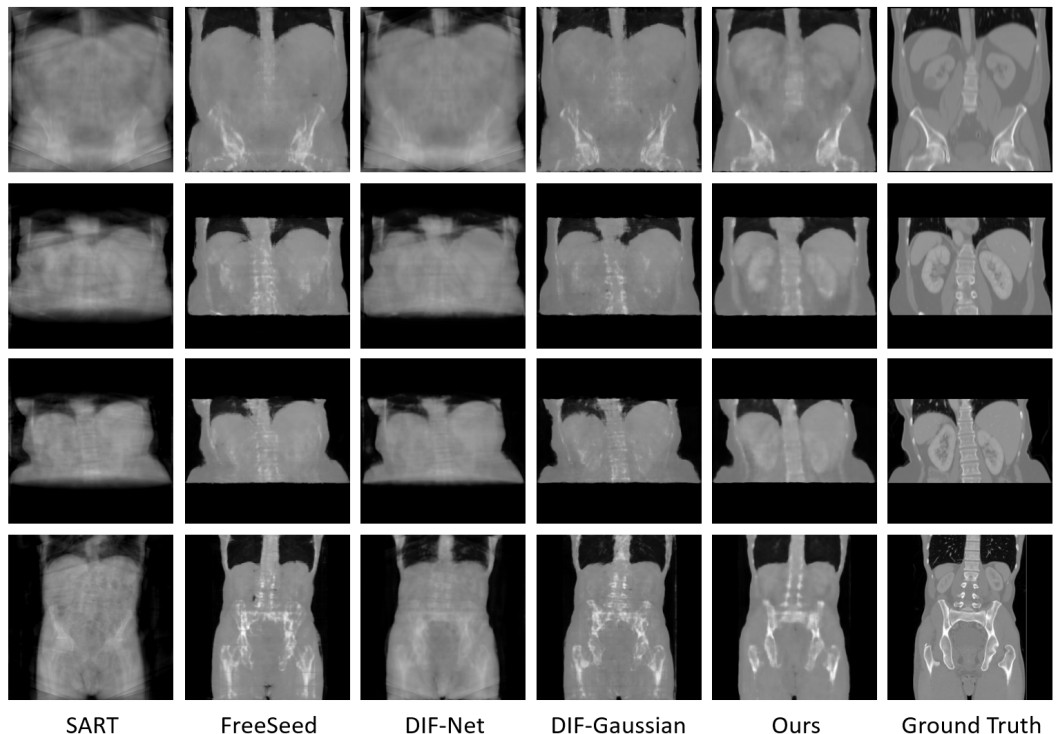

| SART | FreeSeed | DIF-Net | DIF-Gaussian | Ours | Ground Truth |

Figure 6: Qualitative comparison of reconstructed slices with traditional and feedforward models.

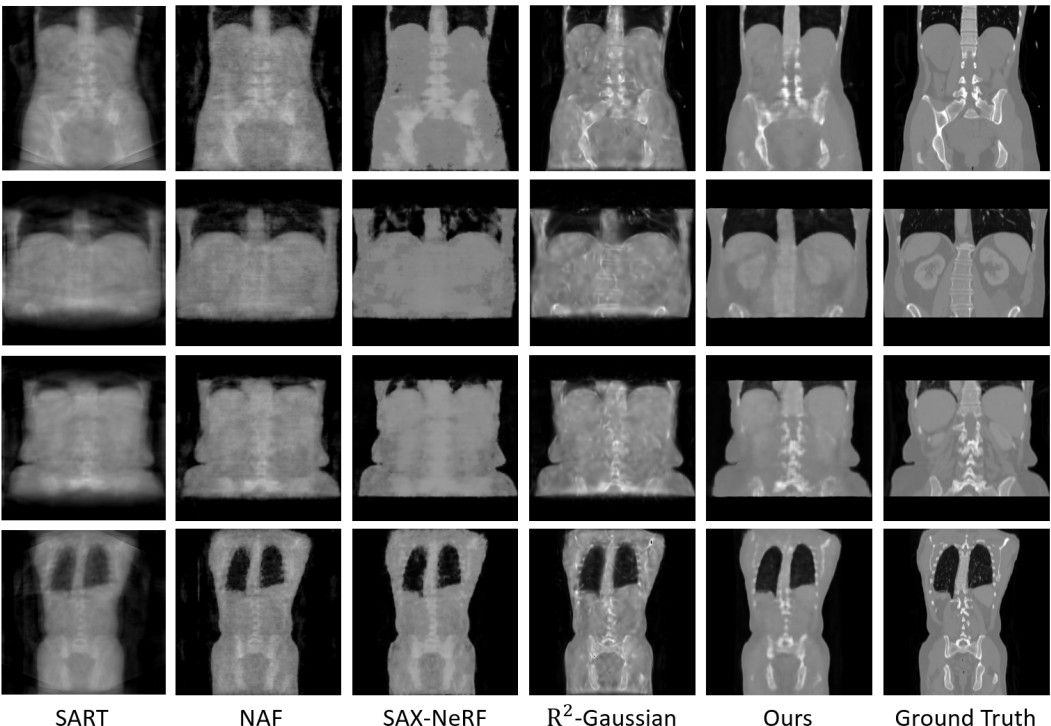

| SART | NAF | SAX-NeRF | $R^2$-Gaussian | Ours | Ground Truth |

Figure 7: Qualitative comparison of reconstructed slices with self-supervised models.

## A.4 Limitations

X-GRM exhibits several limitations that remain future investigation. The model incurs increased memory consumption when utilizing excessive numbers of 3D Gaussians, and demonstrates sub-optimal performance when reconstructing from extremely sparse inputs, particularly single X-ray projections. These challenges in future work through the implementation of efficient Gaussian pruning strategies and the integration of generative modeling approaches, which we leave as future works.

## A.5 Broader impacts

**Impacts on real-world applications.** X-GRM delivers superior reconstruction quality with faster inference speed, enabling efficient clinical deployment. Its accuracy and performance support time-sensitive diagnostic workflows while reducing patient radiation exposure. The VoxGS representation can potentially enable many downstream applications like CT/X-ray registration (as done in DDGS-CT (Gao et al., 2024)), extending the model's utility beyond reconstruction to integrated clinical pipelines.

**Impacts on research community.** This work demonstrates the transformative potential of large-scale data and high-capacity models in medical imaging, encouraging similar approaches in related domains. By open-sourcing code, models, and data, X-GRM contributes essential resources for advancing medical image reconstruction research.

## A.6 More visualization

We provide more qualitative comparisons with traditional, feedforward, and self-supervised methods in Fig. 6 and Fig. 7. Our X-GRM consistently delivers better reconstruction quality with clearer anatomical structures and less noise artifacts.

