# OpenReview forum: "X-GRM: Large Gaussian Reconstruction Model for Sparse-view X-rays to Computed Tomography"
_ICLR.cc/2026/Conference — ICLR 2026 Conference Withdrawn Submission_

### Official Review · Reviewer_J4zm · 2025-10-28

**Soundness:** 3
**Presentation:** 3
**Contribution:** 3
**Rating:** 4
**Confidence:** 3

**Summary:**

The paper presents X-GRM, a large feed-forward Transformer model designed for reconstructing 3D CT volumes from sparse-view 2D X-ray projections. The method introduces two key components: an X-ray Reconstruction Transformer, which employs a per-view ViT encoder and a multi-view fusion ViT conditioned on camera geometry through ModLN for global spatial reasoning, and a novel Voxel-based Gaussian Splatting (VoxGS) representation that places fixed 3D Gaussians at voxel centers to enable both efficient CT volume extraction and differentiable X-ray rendering. The model is trained on nearly 15,000 CT volumes from eight public datasets, leveraging a combination of voxel-wise MSE loss and differentiable rendering constraints. Extensive experiments demonstrate that X-GRM achieves state-of-the-art reconstruction quality, outperforming both traditional iterative and modern feed-forward or self-supervised methods (e.g., FBPConvNet, FreeSeed, DIF-Gaussian, R2-Gaussian, SAX-NeRF) by 3–5 dB PSNR on average, while reconstructing a full 256³ CT volume in less than one second. Furthermore, the approach exhibits strong cross-dataset generalization and supports high-fidelity novel-view X-ray synthesis, highlighting its potential for scalable, real-time, and physically grounded sparse-view CT reconstruction in clinical settings.

**Strengths:**

1. Introducing a feed-forward paradigm for CT reconstruction — The paper extends the recent trend of large reconstruction models (e.g., LRM, GRM) into the medical imaging domain, proposing a purely feed-forward architecture for sparse-view CT that avoids per-case optimization and makes a conceptual step toward scalable “foundation models” for tomographic reconstruction.

2. High efficiency with competitive quality — The proposed model achieves state-of-the-art reconstruction performance while drastically reducing inference time to around one second per 256³ CT volume, demonstrating strong potential for real-time or clinical use cases where traditional iterative or diffusion-based methods are prohibitively slow.

3. Clear and accessible presentation — The paper is well organized and clearly written, with intuitive figures and straightforward technical explanations. The method’s motivation, architecture, and experiments are easy to follow even for readers outside the immediate subfield of CT reconstruction.

**Weaknesses:**

1. Limited realism and generalization — All training X-rays are synthetically rendered from CT volumes using the TIGRE simulator. The absence of experiments on real clinical radiographs makes it unclear how well the model generalizes to real-world acquisition noise, scanner geometry variations, or patient-specific artifacts.

2. Shallow experimental analysis — Although the paper reports many quantitative results, the ablation study remains minimal. Only three components (ModLN, VoxGS, and attention) are tested, while other key factors such as model size, number of input views, and fusion depth are not analyzed, leaving some architectural choices insufficiently justified.

**Questions:**

Real-data generalization: Since all training X-rays are synthetically generated from CT volumes using TIGRE, could the authors provide any preliminary evidence (even qualitative) that the model can generalize to real clinical X-rays with realistic noise and device geometry? Would fine-tuning on a small set of real X-rays be feasible?

1. Role of differentiable rendering: The paper emphasizes that VoxGS enables differentiable X-ray rendering, but the experiments do not clearly show how this component contributes to training or performance. Could the authors include an ablation or visualization that quantifies the benefit of the rendering loss?

2. Architecture scaling and efficiency: How does performance and runtime scale with model size (e.g., ViT-B vs. ViT-L) and number of input views? Are there diminishing returns when adding more views or layers?

3. Cross-dataset transfer: The cross-dataset test on PENGWIN and FUMPE is a good start—could the authors comment on how domain gaps (e.g., anatomy, scanner protocols) affect reconstruction? Is domain adaptation necessary?

4. Clinical interpretability and safety: Have the authors analyzed failure cases or potential clinical artifacts (e.g., missing fine structures, false positives in dense tissues)? Some discussion on interpretability or uncertainty estimation would strengthen the paper’s practical relevance.

---

### Official Review · Reviewer_pLGa · 2025-10-28

**Soundness:** 2
**Presentation:** 3
**Contribution:** 2
**Rating:** 2
**Confidence:** 4

**Summary:**

This paper introduces X-GRM, a feed-forward model for reconstructing 3D CT volumes from sparse-view 2D X-ray projections. It aims to address limitations of existing methods, such as the slow inference of optimization-based approaches and the model capacity or representation constraints of other feed-forward models. The method has two primary components: an X-ray Reconstruction Transformer for encoding and fusing image features from multiple views, and a volume representation termed Voxel-based Gaussian Splatting (VoxGS). In this representation, 3D Gaussian positions are fixed at voxel centers, and the network predicts other attributes like opacity, scale, and rotation. This design allows for CT volume extraction through direct indexing and supports differentiable X-ray rendering, which is used to apply a rendering constraint during training. The authors report that X-GRM, when trained on a large dataset, demonstrated improved reconstruction quality and faster inference times (under one second) compared to several traditional, feed-forward, and self-supervised methods on their test set.

**Strengths:**

1. The writing of the paper is satisfactory.
2. The usage of Voxel-based Gaussian is interesting and supports fast rendering and inference.

**Weaknesses:**

1. The biggest concern I have is about the evaluations of this paper. The paper claims it creates a large-scale dataset following another paper, X-LRM. As most training/inference data are the same across two papers when checking them, I notice the scores reported for the same baseline using same evaluation metric has large discrepancy. The scores reported for baselines in this paper is on average 3-4db (15-20%) lower than those reported in X-LRM. This raises my concern if baseline evaluations are fair in this paper.

2. For baselines included in this paper, it lacks similar baselines that focus on sparse-view CT reconstruction. Most baselines included in this paper are not directly applied to sparse-view task. Additional baselines focused on this specific task should be included.

3. The paper's findings are based exclusively on simulated data. A key potential benefit of this large-scale model is its ability to generalize to data from different real-world scanners, but this practical application remains unevaluated by the authors.

4. The paper's exclusive evaluation in ultra-sparse settings, using 10 views or fewer, limits its applicability to real-world clinical scenarios. As shown in fig3,4,6,7 of the paper, the reconstruction quality is poor and far way off from real-world usage. Consequently, the method's utility for actual diagnostic or surgical guidance remains unevaluated and unclear.

**Questions:**

Please refer to weakness section.

---

### Official Review · Reviewer_1buq · 2025-10-30

**Soundness:** 3
**Presentation:** 4
**Contribution:** 3
**Rating:** 4
**Confidence:** 5

**Summary:**

The manuscript introduces X-GRM, a feed-forward transformer that reconstructs 3D CT volumes from sparse-view X-ray projections in a single pass. The method tokenizes each projection with an encoder ViT and fuses cross-view information with a multi-layer fusion ViT conditioned by Plücker ray embeddings. Decoding uses a new **Voxel-based Gaussian Splatting (VoxGS)** representation with fixed Gaussian centers at voxel locations, predicting opacity/scale/rotation to enable both fast volume extraction and differentiable X-ray rendering. Training jointly matches ground-truth volumes and rendered projections via MSE and weighted L1/SSIM losses. Experiments report higher PSNR/SSIM and faster runtime than traditional and feed-forward baselines, competitive advantages over self-supervised methods, cross-dataset generalization, and strong novel-view synthesis.

**Strengths:**

- **Advanced architectural design.**
  The paper effectively leverages recent advances in natural image modeling while addressing the unique challenges of CBCT reconstruction. The proposed *Scalable Cross-View Transformer Architecture* integrates a strong DiNO-based backbone for feature extraction and adopts *ModLN-conditioned Plücker rays* instead of simple attention-based camera embeddings. Adapting such advanced 3D vision techniques to medical CT reconstruction is a valuable contribution to the community.

- **Clear writing and implementation details.**
  The manuscript is clearly written, with detailed explanations of implementation procedures that substantially enhance reproducibility and transparency.

**Weaknesses:**

- **Missing key feed-forward baseline.**
  Among feed-forward methods, the comparison omits the current SOTA approach **C²RV [1]**. The strongest baseline reported is DIF-Gaussian, which no longer represents the latest standard. I strongly recommend adding **C²RV** for a more complete comparison.

- **Limited evaluation metrics.**
  The paper evaluates reconstruction quality only with PSNR and SSIM. While X-GRM shows clear quantitative gains, qualitative results remain visually blurred. For ultra-sparse (6–10 view) CT scenarios, perfect reconstruction is unrealistic—these settings are often used for fast intra-operative localization. The authors are encouraged to include richer evaluation metrics such as downstream segmentation performance or key-organ volume estimation to better demonstrate practical utility.

- **Insufficient qualitative analysis for cross-dataset generalization.**
  Section 4.3 provides only quantitative metrics, without qualitative visualizations. Moreover, X-GRM’s numerical results appear comparable to DIF-Gaussian and R²-Gaussian, which does not convincingly demonstrate superior generalization.

- **Limited validation of generalization claims.**
  The generalization analysis is restricted to two OOD datasets, with marginal performance improvement. To substantiate the generalization claim, it would be valuable to include results under different CT acquisition geometries.


## References

> [1] Lin, Yiqun, et al. **"C²RV: Cross-Regional and Cross-View Learning for Sparse-View CBCT Reconstruction."** *CVPR*, 2024.

**Questions:**

- **On differentiable X-ray rendering and the motivation for VoxGS:**
  The paper argues that recent methods such as X-LRM and DeepSparse cannot incorporate differentiable X-ray constraints. I have two related questions:
  1. If a neural radiance field is used, why is differentiable X-ray rendering not supported? Additionally, with voxel-based representations and differentiable CT forward models (e.g., *torch-radon*, *TIGRE*, *DiffDRR*), such constraints should, in principle, be implementable.
  2. According to the description, the 3D Gaussians in VoxGS are positioned on fixed voxel-grid coordinates, seemingly sacrificing adaptivity and spatial flexibility while incurring large memory costs. For instance, with a 256³ grid, does the model instantiate 256³ fixed-position Gaussians? How does this differ conceptually from a standard voxel representation? It seems the only potential gain lies in rasterization-based rendering speed—how significant is this improvement? The design appears to constrain the Gaussian representation to an extent that undermines its core advantages.

- How long does the model take to train?
- In Sec. 4.4 (X-ray synthesis), could the authors compare performance against self-supervised approaches on OOD datasets or under varying geometries?
- Could additional qualitative visualizations be provided, especially for different anatomical regions (e.g., head)?

---

### Official Review · Reviewer_473U · 2025-10-30

**Soundness:** 2
**Presentation:** 3
**Contribution:** 2
**Rating:** 2
**Confidence:** 5

**Summary:**

This work proposes an X-ray Gaussian Reconstruction Method (X-GRM), a large supervised model for 3D sparse-view CT imaging. The proposed X-GRM consists of a scalable, pre-trainable ViT and VoxGS, a 3D Gaussian Splatting-based module. The ViT extracts features from input 2D projections, and VoxGS transforms these features into a 3D Gaussian Splatting representation. Finally, through voxelization and rendering, the model is optimized using losses defined in both the volume and projection domains. The proposed X-GRM is a feedforward model, enabling real-time 3D CT reconstruction. The experiments demonstrate its superiority over existing CT reconstruction methods (including both supervised and self-supervised ones) in terms of both running time and image quality.

**Strengths:**

- The paper addresses an important problem, real-time reconstruction for 3D CT imaging, with a clear and meaningful motivation.
- The proposed X-GRM is generally reasonable. Specifically, the ability of ViT to extract rich features from projection data has been well validated in prior works, and the 3D Gaussian Splatting framework indeed shows advantages in high-fidelity representation and real-time rendering. Thus, combining the two is technically sound.
- This work collects a large-scale dataset and conducts comprehensive experiments, which enhances the reliability of the proposed method.
- The paper is clearly written and easy to follow.

**Weaknesses:**

- The motivation for using 3D GS to preserve high data fidelity is somewhat unclear. The paper claims to “support differentiable real-time rendering,” and thus employs 3D GS as a representation to incorporate X-ray physics. However, grid-based representations can also incorporate the X-ray forward model through ray-based rendering. Moreover, real-time rendering is not necessarily critical during model training. Please clarify this motivation.

- In line 232, the paper claims that “complex trilinear interpolation strategies are not required for identifying voxel values; instead, we can directly obtain them from Gaussian opacities.” However, in my opinion, trilinear interpolation is already quite simple. Please clarify what is meant here.

- The main difference between VoxGS and the original 3D GS is that VoxGS uses fixed Gaussian positions (i.e., means). In my opinion, this design choice is questionable. One major reason 3D GS performs well is its ability to adaptively adjust positions, scales, and shapes. Fixing the positions may significantly limit this adaptive capability.

- From the experimental results, although the proposed method outperforms the baselines, the reconstructed images still appear overly smooth and lack fine structural details, which may not meet the requirements of clinical diagnosis.

**Questions:**

See Weaknesses, please.

---

### Note · Authors · 2025-12-03

I have read and agree with the venue's withdrawal policy on behalf of myself and my co-authors.